# Combined Transcriptomic and Metabolomic Approach Revealed a Relationship between Light Control, Photoprotective Pigments, and Lipid Biosynthesis in Olives

**DOI:** 10.3390/ijms241914448

**Published:** 2023-09-22

**Authors:** Tiziana Maria Sirangelo, Ivano Forgione, Samanta Zelasco, Cinzia Benincasa, Enzo Perri, Elisa Vendramin, Federica Angilè, Francesco Paolo Fanizzi, Francesco Sunseri, Amelia Salimonti, Fabrizio Carbone

**Affiliations:** 1Research Centre for Olive, Fruit and Citrus Crops, Council for Agricultural Research and Economics (CREA), Via Settimio Severo, 83, 87036 Rende, Italy; 2Research Centre for Olive, Fruit and Citrus Crops, Council for Agricultural Research and Economics (CREA), Via di Fioranello, 52, 00134 Rome, Italy; 3Department of Biological and Environmental Sciences and Technologies, University of Salento, Via Lecce-Monteroni, 73100 Lecce, Italy; 4Department Agraria, University Mediterranea of Reggio Calabria, Località Feo di Vito, 89124 Reggio Calabria, Italy

**Keywords:** light signal transduction, olive ripening, gene expression analysis, targeted RNA-Seq, HPLC-MS/MS

## Abstract

Olive possesses excellent nutritional and economic values for its main healthy products. Among them, a high content of antioxidant compounds, balanced during the ripening process, are produced under genetic and environmental control, resulting in high variability among cultivars. The genes involved in these complex pathways are mainly known, but despite many studies which indicated the key role of light quality and quantity for the synthesis of many metabolites in plants, limited information on these topics is available in olive. We carried out a targeted gene expression profiling in three olive cultivars, Cellina di Nardò, Ruveia, and Salella, which were selected for their contrasting oleic acid and phenolic content. The –*omics* combined approach revealed a direct correlation between a higher expression of the main flavonoid genes and the high content of these metabolites in ‘Cellina di Nardò’. Furthermore, it confirmed the key role of *FAD2-2* in the linoleic acid biosynthesis. More interestingly, in all the comparisons, a co-regulation of genes involved in photoperception and circadian clock machinery suggests a key role of light in orchestrating the regulation of these pathways in olive. Therefore, the identified genes in our analyses might represent a useful tool to support olive breeding, although further investigations are needed.

## 1. Introduction

Olive oil represents a very important product in Mediterranean countries, whose beneficial properties are widely recognized. A diet promoting the consumption of olive oil has proved highly effective to reduce cardiovascular diseases and different kinds of cancer; further, olive oil assumption can regulate cholesterol metabolism, playing a key preventive role for human health [1].

The biochemical events occurring during olive ripening resulted in important changes affecting the oil content and quality, including the fatty acids (FAs) composition and the phenolic profile with a consequent accumulation of anthocyanin pigments instead of other phenols [2].

Phenols represent the largest group of plant secondary metabolites, with their significantly variation in size from one aromatic ring to complex structures such as lignin [3]. Flavonoids are the largest group of naturally occurring beneficial phenolic compounds, acting as antioxidants as protectants against cardiovascular and other human diseases. Phenolic compounds in olive are mainly represented by phenolic alcohols, secoiridoids, and lignans that have attracted researchers for their interesting biological properties [4]. Furthermore, they are responsible for most sensory properties of olive oil, such as its peculiar bitter and spicy taste [5].

The phenol biosynthesis in olive fruits during ripening was deeper characterized [6,7], although the biosynthetic pathway of secoiridoids has been only partially elucidated [8].

In olive, phenylpropanoids are abundant in active substances, such as luteolin, apigenin, caffeic acid, ferulic acid, and others [7]. In their biosynthesis, the first step is the p-coumarate-CoA formation from phenylalanine, and the second step includes the chalcone biosynthesis through three and one molecules of malonyl-CoA and p-coumarate-CoA, respectively. Finally, naringenin, which is crucial for the synthesis of several flavonoids, is synthesized [9].

FAs are chemical species detected in relevant amounts in olive oil, representing the major constituent of triacylglycerols (TAGs). In olives, they are produced and accumulated mainly in the pericarp (>90%) rather than in the seed. They are mainly composed by the monounsaturated oleic acid (C18:1; ~50–83% of all the TAGs) and the linoleic acid (C18:2; ~3.5–21%), followed by palmitic (C16:0; 7.5–20%), stearic (C18:0; 0.5–5%), and linolenic acids (C18:3; ~0–1.5%) [10,11]. The olive oils with high oleic and low linoleic, linolenic, and palmitic acids’ content are considered better for the nutritional and technological values [12,13].

The plant biosynthesis of olive oil healthy compounds, such as FAs and polyphenols, is influenced by several external factors including light exposure, altitude, latitude, temperature, rainfall, cropping management, and post-harvest processing, but the genotype is considered decisive for their content variability [14,15].

Many studies focused on the relationship between radiation and oil biosynthesis, demonstrating the role of limited irradiance in the oil quality [16,17]. Indeed, it is well known that plants have adapted their growth and development to the diurnal light/dark cycling, and the diurnal rhythm of the genes’ expression is achieved via light and a free-running internal circadian clock. In particular, *Arabidopsis* genes encoding for phytochrome B, cryptochrome 1 and 2, and phototropins were extensively reported as clock-regulated and the circadian clock may orchestrate the production of photoprotective pigments early in the day via the phenylpropanoids’ biosynthesis of the genes’ peak [9]. Some MYB transcription factors, which are involved in several biological processes, belonged to the CCA1-like (circadian clock-associated 1) group, including several genes that are key regulators of the circadian clocks [11]. These genes regulate phenylpropanoid metabolism by activating the structural genes involved in the anthocyanin pathway in many crops [12,13].

Nowadays, a genome reference sequence may provide a platform for isolating and functionally characterizing genes via integrated approaches (transcriptomics, proteomics, and functional studies). In the last decade, the olive genome first draft, a platform for the genes’ characterization, and next generation sequencing (NGS) techniques’ widespread use contributed to the advance of olive tree genomics knowledge [4,10,18,19,20].

The integrated analysis of FA metabolism and the transcriptome performed at different development stages as well as a comparative transcriptome investigation on the drupes collected from plants growing in areas at different altitudes demonstrated the differential accumulation of olive quality compounds [11,21]. Moreover, comparative transcripts analyses were able to identify relevant differentially expressed genes in regulating the olive phenols’ and tocopherols’ content during ripening [8,22,23]. The decrease in the total phenolic content (TPC) correlated to the expression pattern of six genes included in the pathway, along with the accumulation of total flavonoids and anthocyanins which were also observed [24,25,26]. Furthermore, the molecular relationship between the flavonoids’ biosynthesis and circadian cycling was recently reported by using transcriptomic approaches. In detail, the phytochromes (red/far-red light-absorbing receptors) and the cryptochromes (blue light and UV-A photoreceptors) showed a mediating role for plant growth, development, and ripening [27,28]. Otherwise, the expression level of a large number of genes is affected by light, and this effect may differ in different crop genotypes [29].

Here, a panel of known genes was defined for elucidating the relationships among the expression pattern of the genes’ network involved in the response to light and photoperception and those involved in the biosynthesis of primary (TAGs) and secondary (phenylpropanoids) metabolites in olive.

Interestingly, our study highlighted the involvement of photoreceptors and circadian clock machinery in the production of photoprotective pigments and lipid modification in the olive tree. The combined transcriptomics and metabolomics approach allowed us to elucidate the role of genetic and environmental factors on these pathways during ripening, revealing the differences on polyphenols and oleic acid accumulation among cultivars.

## 2. Results

To analyze the expression level of 257 gene targets and the differential accumulation of 29 metabolites in three different olive cultivars (‘Cellina di Nardò–CdN; ‘Ruveia’–Ru; and ‘Salella’–Sa), contrasting for their different oleic acid as well as different phenolic compounds contents, a combined approach based on targeted RNA sequencing, HPLC-MS/MS and GC, was adopted.

The three contrasting cultivars were selected based on a preliminary screening during the 2018 olive oil season from an extensive number of cultivars included in the CREA germplasm collection via ^1^H NMR spectroscopy analysis of oleic, linoleic acids, tyrosol, hydroxytyrosol, and their derivatives’ content. Only the data acquired for the three selected cultivars are reported in Table 1.

In detail, the analysis was carried out on 50 fruit mesocarps from the three plants of each cultivar sampled at the Turning (T-Jaèn index, between 2.0 and 2.5) and Ripe (R-11 days after T) stages during the 2019 olive oil season (Appendix A). On the same samples, metabolite analyses to determine the polyphenols’ content and the fatty acids’ (FAs) concentration were performed.

### 2.1. Targeted RNA-Seq Analysis

The panel of gene targets was designed on the annotated genome of the Spanish cultivar (cv.) Farga, Version 6 [18], the only available release in 2018 for cultivated olive, when the gene panel was designed and produced, including 48 and 112 genes involved in the phenylpropanoid pathway and the fatty acid metabolism, respectively. Moreover, the panel included 97 additional genes playing a role in the photoperception and the circadian clock machinery, with key regulatory functions in the biosynthesis of secondary metabolites (Appendix A).

The high-quality filtered reads were mapped to the assembled olive reference genome version 6 [18], keeping the same gene IDs of the panel to facilitate the understanding, in order to identify the differentially expressed genes (DEGs) between the genotype pairs and the ripening stages. The sequenced and described genes have been reannotated; therefore, in all the tables and figures are reported fully updated functional annotations for each gene via BLAST analysis. The alignments’ percentage of the mapped reads on the genome reference was comparable within the replicates and cultivars (Appendix A).

To identify the genes directly involved in fruit ripening and the regulation of fatty acids, as well as in the polyphenols synthesized via the phenylpropanoid pathways, two approaches were adopted. First, the comparisons between the T and R stages for each genotype (CdN-T vs. R, Ru-T vs. R, and Sa-T vs. R) were carried out; then, we compared the genotypes with contrasting fatty acid and polyphenol contents at the same ripening stage (CdN vs. Sa and Ru vs. Sa).

A principal coordinate analysis (PCoA) based on the RNA-Seq data revealed a clear segregation of the cultivars according to their metabolite contents (Figure 1). The major component of variance (x axis) significantly contributed to the polyphenols’ variability and was able to distinguish the CdN and Sa cultivars. The y axis showed the variance for FAs, which was able to distinguish the Ru cultivar from the others.

#### 2.1.1. Differential Gene Expression Profiles across Fruit Ripening in Different Cultivars

To identify the genes that were differentially regulated (FDR < 0.05) in each genotype during fruit ripening, a transcript abundance analysis was performed using edgeR. The analysis led to the identification of 75 DEGs by comparing the two stages (T vs. R) in each genotype. Thirteen out of 75 DEGs were shared among the genotypes (Group I, Appendix A; Figure 2A), and 18, 7, and 5 DEGs were shared between CdN and Sa, CdN and Ru, as well as Ru and Sa, respectively (Group II, III, and IV, Appendix A; Figure 2A).

During ripening, an expression trend of the genes related to the decrease in photosynthetic activity, as well as the chloroplast to chromoplast transition, was observed. Indeed, the transcript levels of seven photosystem I (OE6A004297, OE6A028974, OE6A024133, OE6A001158, OE6A076354, OE6A015647, and OE6A050501), two photosystem II (OE6A073013 and OE6A018659) subunits, and the *chlorophyll a–b binding* proteins (OE6A106143, OE6A114109, OE6A092384, OE6A080272, OE6A050501, OE6A066649) significantly declined in at least one of the three cultivars in the T vs. R stages. The same pattern of expression was found for the genes with key functions in plastid genome regulation such as RNA polymerase sigma factor *SIGB* (OE6A012335), *SIGE* (OE6A065557), and *sigC* (OE6A109185), the flowering locus T (*FLT*-OE6A103537), some members of the *CONSTANS-LIKE* Transcription Factors (TFs) family (OE6A043940, OE6A106820, OE6A082516, and OE6A054446), and *HY5* (OE6A086721). In addition, a gene encoding for *EARLY FLOWERING 4-like (ELF4*–OE6A060484) showed a similar regulation (Appendix A).

Finally, among the overexpressed at the R stage, the genes encoding for the flowering locus K (*FLK*-OE6A113194), the transcription factor *PIF1-LIKE* (OE6A089246), and the transcription factor *PIL1-LIKE* (OE6A010456) involved in the light-dependent induction of tocopherol biosynthesis during fruit ripening were found (Appendix A).

*FNSII* (OE6A081156), *UFGT* (OE6A074408), *FLS* (OE6A040780), and *F3′5′H* (OE6A068581)-encoding flavone synthase II, UDP-glucose flavonoid 3-O-glucosyltransferase 3, flavonol synthase, and flavonoid 3′,5′ -hydroxylase, relevant enzymes involved in the flavanols’ biosynthesis and the glycosylation of anthocyanins, showed a decreasing expression at the R stage in at least two out of the three cultivars (Appendix A). Interestingly, most of the genes encoding enzymes involved in the phenylpropanoid biosynthesis, such as dihydroflavonol 4-reductase (*DFR*-OE6A104547) and naringenin, 2-oxoglutarate 3-dioxygenase (*F3H*-OE6A005638) were over-expressed in CdN and Ru; trans-cinnamate 4-monooxygenase (*C4H*-OE6A108606) was overexpressed in Ru and Sa; and the leucoanthocyanidin dioxygenase (*LDOX/ANS*-OE6A081102 and OE6A048749), phenylalanine ammonia-lyase (*PAL*-OE6A048764), and trans-resveratrol di-O-methyltransferase-like (*OMT*-OE6A065332), were overexpressed in all three cultivars at the R stages (Appendix A).

In the T vs. R stages, different expression patterns among genotypes in the genes involved in the FA biosynthesis were also observed (Appendix A). Among others, some over-expressed genes encoding for the members of the 3-ketoacyl-CoA synthase family (OE6A013923, OE6A090531, OE6A044469, and OE6A048706) in the early ripening stage in the genotype with the lowest oleic acid content (Sa) were found; by contrast, two members of the same gene family (OE6A013923 and OE6A012604) resulted in the down-regulation of the genotype with the highest oleic acid content (Ru).

Moreover, the transcripts encoding for two very-long-chain enoyl reductase-like (OE6A060413 and OE6A077716) were found up-regulated at the T stage, while *MCAT* (malonyl-[acyl-carrier-protein] transacylase-OE6A056459), *ACCB* (biotin carboxyl carrier of acetyl-carboxylase-OE6A112905), and *SAD* (stearoyl-[acyl-carrier-protein] 9-desaturase-OE6A002165) transcripts resulted in down-regulation at the same stage. Key genes involved in TAG biosynthesis, such as *PDAT 1* and *2* (phospholipid: diacylglycerol acyltransferase-OE6A078092, OE6A001246) and *PDCT1* (phosphatidylcholine:diacylglycerol cholinephosphotransferase 1-like-OE6A057041), were differentially expressed in the ripening stages. For instance, the transcripts were up-regulated during the ripening only in one cultivar (Sa or CdN) (Appendix A).

#### 2.1.2. Comparisons between Cultivars with Contrasting Polyphenols’ Content

To increase the insights into the functions of genes related to the phenylpropanoids’ biosynthesis in olive, a comparative analysis of transcript abundances was performed between genotypes with contrasting polyphenol contents CdN and Sa that showed a high and low content, respectively (Table 1). The analysis revealed 90 DEGs in at least one of the ripening stages, with a half of these genes overexpressed in CdN (Appendix A). Most of the 90 DEGs (53) were shared by both cultivars (group I, Appendix A; Figure 2B), regardless of the ripening stage.

Many putative light-regulated genes involved in photoperception, photosynthetic machinery, and circadian rhythm were differentially regulated in CdN vs. Sa (Appendix A). In detail, the genes that regulate the anthocyanin biosynthesis belong to different families of far-red and blue light receptors, such as phytochrome A (*PHYA*-OE6A074596), far-red impaired response 1 (*FAR1*-OE6A012821 and OE6A091010), far-red elongated hypocotyl 3 (*FHY3*-OE6A044285), cryptochrome-1 (*CRY1*-OE6A114272 and OE6A036399), and a member of the *PIL* transcription factor family *(PIF3*-OE6A010456) were coherently overexpressed in CdN (Appendix A). *PHYA* and *CRY1* were also involved in the upstream regulation of the circadian clock factors via key genes such as *SPA1* (OE6A083912), *LHY* (OE6A024312), and *REVEILLE 6* (OE6A044599), some members of *CONSTANS-LIKE* TFs (*COL 10*-OE6A114181 and *COL 16*-OE6A054446), the transcription factor *PIF4* (OE6A078915), a repressor of circadian clock elements *LOV DOMAIN CONTAINING* gene (OE6A073435), and a RNA polymerase sigma factor *SIGC* (OE6A109185 and OE6A092574) that appeared down-regulated in CdN (Appendix A).

Interestingly, an up-regulation of *EARLY FLOWERING 3-LIKE* transcript gene (*ELF3*-OE6A047144), *CONSTANS 1* and *2* (*CO1*-OE6A035887 and *CO2*-OE6A082516) promoting the *FT* gene expression (OE6A103537), photosystem I and II subunits (*PSAK*-OE6A028974, *PSAF*-OE6A015647, *PSB-LIKE* OE6A073013, and *PSAN*-OE6A024133) and the chlorophyll a-b binding members (*CAB8*-OE6A080272, *LHCB5*-OE6A066649, and *CAB-LIKE*-OE6A092384) in CdN was observed (Appendix A).

As expected, in general, the flavonoid pathway genes leading to the anthocyanin formation were strongly overexpressed in CdN in agreement to the higher p-coumaric acid, caffeic acid, luteolin, rutin, and anthocyanin content measured in CdN fruits (Figure 3; Appendix A).

On the contrary, the genes related to the accumulation of epicatechins, *ANR* (anthocyanidin reductase-OE6A113723) and the promoting 3′-5′-hydroxylation of flavan-3-ols (*F3′5′H*-OE6A055813) resulted in the up-regulation in Sa (Appendix A, Figure 3).

#### 2.1.3. Comparisons between Cultivars with Contrasting Oleic Acid Content

Ninety-three gene transcripts showed a different expression profile in the Ru vs. Sa at least in one ripening stage (Figure 2C, Appendix A). The number of up- and downregulated gene transcripts was equally distributed, with 58 out of 93 DEGs shared by both cultivars, regardless of the ripening stage (Figure 2C).

The genes involved in the phenylpropanoids’ biosynthesis, photoperception, and photosynthetic machinery appeared differentially expressed between cultivars (Appendix A). The cultivar Ru showed higher levels of phenolic compounds compared to Sa, even though it had more limited differences (Table 1). A coordinated regulation of key genes in the cascade of light-mediated processes was also observed. Interestingly, an opposite expression trend, with respect to the comparison CdN vs. Sa, was here observed with the key regulatory gene in the anthocyanin biosynthesis PIL Transcription factor (*PIF3*-OE6A010456) and the up- and down-regulation at the T and R stages, respectively. Anyhow, many genes in the phenylpropanoid pathway (*PAL*-OE6A095147, OE6A049944, OE6A048764, *DFR*-OE6A082229, *FLS*-OE6A040780, *FLS2*-OE6A081156, *F3GT*-OE6A074408, OE6A053999, *OMT*-OE6A065332, OE6A106272, and *LAR* OE6A011825) resulted in the up-regulation in Ru compared to Sa (Appendix A).

Interestingly, the *FT* (OE6A103537) transcript levels were strongly induced by the concurrent *CRY1* (OE6A036399), *PHYA* (OE6A114644), *FAR1* (OE6A012821), and *CO1* (OE6A035887) up-regulation and the *SPA1* (OE6A045568), *COL10* (OE6A114181), and *FLOWERING LOCUS D* (*FD*-OE6A008716), as well as the *LOV DOMAIN-CONTAINING* (OE6A07394, OE6A105055, and OE6A007397) and the *KELCH DOMAIN-CONTAING 2-LIKE* (OE6A084581) down-regulation (Appendix A).

Finally, the photosystem I and II subunits (*PSA3*-OE6A004297, OE6A076354, *PSAF*-OE6A015647, *PSAK*-OE6A028974, *PSB-LIKE*-OE6A073013, and *PSBN*-OE6A024133) and the chlorophyll a-b binding members (*CAB6*-OE6A050501, *CAB8*-OE6A080272, *CAB-CP24*-OE6A114109, *CAB-CP26*-OE6A066649, *CAB-151*-OE6A106143, and *CAB-like*-OE6A092384) up-regulation were observed in Ru compared to Sa (Appendix A).

The up-regulated transcripts in Ru vs. Sa shared between the ripening stages were related to the genes involved in the FA and TAG biosynthesis pathways, which were encoding for the acetyl-CoA carboxylase biotin carboxyl carrier protein (OE6A115934, OE6A092496), the 3-oxoacyl-[acyl-carrier-protein] synthase III (OE6A034993), the 3-ketoacyl-CoA synthase (OE6A013923, OE6A090531, OE6A044469, OE6A042315, and OE6A029462), and the very-long-chain enoyl-CoA reductase (OE6A077716 and OE6A065720) (Appendix A, Figure 4).

On the other hand, two *FAD2-2* gene transcripts encoding for a delta (12)-fatty-acid desaturase (OE6A051290 and OE6A011870) were down-regulated in Ru compared to Sa (Appendix A, Figure 4).

### 2.2. Combined Transcripts and Metabolites Analysis of DAMs and DEGs

The levels of different flavonoid and phenolic compounds in the comparison between CdN and Sa, contrasting for the polyphenols’ content, were determined on the same samples used for the RNA-seq experiment (Figure 5, Appendix A).

Many compounds showed significant genotype-dependent differences at each ripening stage (Figure 5). Nine out of eighteen metabolites were strongly more accumulated in CdN compared to Sa in both ripening stages (Figure 5A), while higher levels of the further five metabolites were observed in the ripe drupes of CdN (Figure 5B). The other metabolites showed an accumulation pattern mediated via the ripening processes and three of them were more expressed in CdN (Figure 5C,D). Moreover, significantly higher levels of anthocyanins have been observed in CdN at both ripening stages compared to the other two cultivars, as well as a significant increase in these metabolites during ripening in each genotype (Appendix A).

According to the threshold of Pearson’s correlation coefficient (PCC) > |0.90|, among 19 Differential Accumulated Metabolites (DAMs) (Figure 5) and 90 significant DEGs (Appendix A), fifty-four significant DEGs correlated to the DAMs were identified (Appendix A). The correlation analysis showed that each metabolite was closely related to many genes (Appendix A).

A similar approach was used to determine the FAs’ composition (%) via GC in the oils between Ru and Sa, contrasting for the oleic acid content. In particular, the analysis was performed on the oils from drupes collected at both ripening stages, T and R, providing the ratios of thirteen FAs (Figure 6). The highest oleic acid percentage was observed in the oils from Ru at both the turning and ripe stages, as expected (about 73%, Figure 6B). The oleic acid content increase in Ru was balanced by the significantly lower levels of linoleic acid (8%) compared to 17% measured in Sa (Figure 6A). A similar trend was observed for palmitic acid with a significant ratio difference of about 4% between the genotypes (Figure 6A), while the eicosenoic acid (C20:1), behenic acid (C22:0), and the lignoceric acid (C24:0) levels differed significantly between the genotypes, mainly in the oil from the ripe drupes of Ru (Figure 6B). The remaining four DAMs did not show a particular expression pattern or showed a similar trend to the other two groups (Figure 6C). Finally, the only three FAs did not show a differential expression (Figure 6D).

According to the threshold of PCC > |0.90|, between 10 DAMs (Figure 6) and 93 significant DEGs (Appendix A), forty-nine significant DEGs correlated to the DAMs were identified (Appendix A). The correlation analysis showed that each metabolite was closely related to many genes. In particular, the highest number of DEGs is related to the oleic, linoleic, and palmitic acids according to their level of accumulation (Appendix A).

## 3. Discussion

A combined transcriptomics and metabolomics approach was useful to shed light on the genetic regulation mechanisms mediated via light and related to the photoprotective pigments and lipid biosynthesis in olive. A correlation between the levels of gene transcripts and the related metabolite accumulation was observed for both the phenylpropanoids’ and FAs’ biosynthesis. PCA was able to distinguish the genotypes according to their polyphenols’ and FAs’ content; otherwise, the drupes from all the genotypes clustered for the T and R stages, confirming that the last step of the fruit ripening is crucial for the synthesis of these metabolites. Our results confirmed the coordinate expression of the genes involved in the FAs’ and phenylpropanoids’ biosynthesis, photoperception, and circadian machinery during the fruit ripening, as previously reported with some differences among genotypes [11,22,32,33,34,35,36,37,38,39].

In agreement to Alagna et al. [22], a significant down-regulation of genes involved in the flavanols’ biosynthesis in all the genotypes during the drupes’ maturation from the T to R stage was observed, mainly in CdN and Ru, sustaining their ability to accumulate these compounds in the early stage of ripening.

Many of the FAs’ biosynthesis genes among the differentially expressed during ripening was more expressed at the R stage in at least one of the three genotypes, in agreement with their accumulation pattern in olive fruits [22,34]. Finally, across the two late maturity stages (T and R), the fatty acid composition (%) was not different, as previously reported [40].

The comparison between contrasting genotypes for the polyphenols’ content highlighted the key role of genes involved in the biosynthesis of anthocyanins, p-coumaric acid, caffeic acid, luteolin, and rutin, as previously reported [8,11,22,25]. Interestingly, the flavonoids, rutin, and luteolin 7-O-glucoside were reported to increase during olive fruit ripening [41] as well as p-coumaric acid and caffeic acid [42]. Moreover, significant differences in the flavonoid and anthocyanin transcripts during fruit maturation were also described in a genotype characterized by a switch-off in skin color [25] as well as in plants growing at different altitudes [11].

All the genes encoding structural enzymes in the pathways of flavonoids resulted as higher expressed in the genotype CdN with a higher content of the main classes of phenols, providing evidences of the crucial role in the transcriptional control of this pathway.

The presence of genes involved in the transcriptional regulation processes mediated via light, among the identified DEGs, confirmed their potential role in the modulation of phenylpropanoid biosynthesis also in the olive tree, suggesting a similar role for the other polyphenols. Interestingly, the decisive role of the metabolic reactions’ cascade induced by the different light spectra in the regulation of important secondary metabolites has been already reported in other plant species [32,43,44,45,46,47].

The *PIF3* and *HY5* TFs’ role to enhance anthocyanin biosynthesis by binding the key gene promoters in *Arabidopsis* under the far-red condition was already reported as well as the negative role of *PIF4* [48,49]. Interestingly, the co-regulation network of anthocyanin biosynthesis observed in CdN sustained its higher red-light perception and determined a further accumulation of these metabolites via the correlated reaction cascade. Indeed, the concurrent *PHYA*, *FHY3*, *FAR1,* and *PIF3* up-regulation and *SPA1* and *PIF4* down-regulation in CdN appeared correlated to the increase in phenylpropanoid biosynthesis mediated via the far-red light. Otherwise, the blue light could also repress the activity of some TFs in the phenylpropanoid pathway via an increased expression of cryptochrome genes. The activation of the reaction cascade mediated via the red and blue light and the related regulatory networks observed in CdN agreed to several evidences from other plants [37,50,51,52,53,54,55,56] (Figure 7).

Flavonoids and non-flavonoid polyphenols are both synthetized by tyrosine. In the olive oil, the main phenols are derivatives from tyrosol and hydroxytyrosol in a larger amount than 90% compared to flavonoids. Unfortunately, limited information about the genes involved in oleuropein and in the other tyrosol and hydroxytyrosol synthesis are available. Otherwise, the known candidate gene set allowed us to take a step forward in the knowledge of the relationship between flavonoids and light, confirming similar trends observed in other plant species and, in addition, highlighting the specific behavior of the olive tree.

Moreover, the anthocyanins are phenylpropanoids responsible for fruit coloration, but their large accumulation did not determine changes in the phenolic composition and the oil’s functional properties. Interestingly, our results highlighted a correlation between the shared elements between flavonoids and non-flavonoid polyphenols and light, confirming similar observations in other plants. Altogether, the largest accumulation of derivates from tyrosol and hydroxytyrosol measured in cv CdN may have been due to a coregulated, light-mediation of most polyphenols. An interest in the olive breeding programs aiming to increase the quality of the oil is also focused on these relationships that will certainly be paid attention to by the scientific community in the next years.

The comparison between the genotypes characterized by the contrasting percentages of oleic and linoleic acid highlighted the role of the differential *FAD2-2* expression between genotypes in determining the different ratios between these two important FAs, as expected. The higher *CRY1* and *PHYA* transcript abundances in the genotype characterized by the highest oleic acid content correlated to the lowest phenylpropanoid levels in Sa. A correlation between the light quality and fatty acids’ regulation seemed to be possible, due to the *FT* repressor role in the transcriptional regulation of some genes belonging to the fatty acid desaturates family, which was reported in many plants [37]. Interestingly, a higher *FT* transcripts abundance in Ru (showing the highest oleic acid content), was mediated by the cryptochromes and phytochromes as well as the concurrent decreased *FD* and *CONSTANS-LIKE 10* expressions, which agreed to the *FAD2-2* down-regulation observed in our experiment. In agreement, the repressor role of *FD, CONSTANS-LIKE,* and *FT* was already reported in *Arabidopsis* and rice [54,55,57]. These results led to the hypothesis of a possible key role of the photoperception reaction cascade in the control of the FAs’ accumulation and ratios (Figure 7).

## 4. Materials and Methods

### 4.1. Plant Material

Three olive cultivars were selected for their different oleic acid as well as phenolic compounds (tyrosol, hydroxytyrosol and their derivatives’ range of 6.74–6.78 ppm) contents. In detail, ‘Cellina di Nardò’ (CdN), ‘Ruveia’ (Ru), and ‘Salella’ (Sa) were chosen for a higher phenol content, oleic acid content, and a lower content of both chemical families, respectively (Table 1). These cultivars were selected in 2018 via a preliminary varietal phenotypic screening conducted using ^1^H NMR spectroscopy analysis on about 100 olive cultivars at three different ripening stages (Green, Turning, and Ripe) from the olive germplasm collection field of the CREA-Research Centre for Olive, Fruit and Citrus Crops (Mirto-Crosia, Cosenza, Italy, 39°37′04.57″ North latitude, 16°45′42.00″ East longitude).

In 2019, a sample of drupes was collected from each selected variety at the turning (T, Jaén index, between 2.0 and 2.5) and ripe stages (R, 11 days after T) and kept at −80 °C until RNA extraction (Appendix A).

### 4.2. Olive Oil Extraction and ^1^H NMR Spectroscopy Analysis

The oils were extracted from the olive samples by using a laboratory scale milling method in a short time, reducing any type of deterioration due to thermal effects, as already reported in literature [58]. Briefly, for each cultivar sample, the olives were plunged into liquid N_2_ and ground to obtain a paste with a stainless-steel blender. Successively, distillated water (2–4 mL) was added to the paste and the obtained mixture was stored over night at 4 °C. The oil was obtained via centrifugation and stored in amber vials until needed for Nuclear Magnetic Resonance analysis.

For NMR analysis, the samples were prepared by dissolving ~140 mg of olive oil in CDCl_3_ in a ratio of olive oil:CDCl_3_ to 13.5:86.5 (%*w*/*w*). Then, 600 μL of the obtained mixture were transferred into a 5 mm NMR tube. The ^1^H NMR spectra were acquired using the Bruker Avance III spectrometer (Bruker Italia, Milan, Italy) operating at 400.13 MHz, T = 300 K, and equipped with a BBI 5 mm inverse detection probe incorporating a z axis gradient coil. NMR experiments were performed under full automation, after loading the individual samples on a Bruker Automatic Sample Changer interfaced with the IconNMR software v3.5 (BioSpin Business Unit, Bruker Italia, Milan, Italy). In order to optimize the NMR conditions, automated tuning and matching, locking and shimming, and a 90° hard pulse calibration P (90°) were carried out for each sample using standard Bruker routines ATMA, LOCK, TOPSHIM, and PULSECAL. Two ^1^H NMR experiments were performed for each sample: the standard one-dimensional, ^1^H ZG, NMR experiment, and the one-dimensional ^1^H NOESYGPPS NMR pulse sequence to enhance the signals of the minor components present in EVOOs. The spectra were obtained via the following conditions: zg pulse program (for ^1^H ZG), 64 K time domain, spectral width of 20.5555 ppm, a receiver gain of 4 and number scans of 16; and noesygpps1d.comp2 pulse program (for ^1^H NOESYGPPS NMR), 32 K time domain, spectral width of 20.5555 ppm, p1 12.63 μs, pl1–1.00 db, 32 repetitions. All the ^1^H NMR spectra were obtained via the Fourier Transformation (FT) of the FID (Free Induction Decay), applying the exponential multiplication with a line broadening factor of 0.3 Hz, which was automatically phased and baseline corrected. Chemical shifts were reported with respect to the TMS (tetramethylsilane as an internal standard of 0.03% *v*/*v*) signal set at 0.00 ppm, obtaining good peak alignment [59].

Fatty acid percentage was calculated according to the procedure already reported in literature [30], where the acquisition parameters of the relative ^1^H ZG NMR experiments were set to the obtained quantitative data (integrals values inaccuracy < 2.0%) [31]. For the minor component, the signals corresponding to the phenolic compounds, in particular, tyrosol, hydroxytyrosol and their derivatives (range of 6.74–6.78 ppm), were integrated and the values from the three selected cultivars are reported in Table 1.

### 4.3. Transcript Analysis

#### 4.3.1. RNA Isolation

Three independent RNA extractions for each genotype and ripening stage, via a RNeasy Plant Mini Kit (Qiagen, Hilden, Germany) according to the manufacturer’s protocol, were performed. To remove DNA contaminations, all the samples were processed with the Invitrogen™ TURBO DNA-free™ Kit (Thermo Fisher Scientific, Waltham, MA, USA).

The nucleic acid purity was analyzed via the Thermo Scientific™ NanoDrop™ 2000c Spectrophotometer (Thermo Fisher Scientific, Waltham, MA, USA) and the samples with 260/280 and 260/230 nm absorbance ratios greater than 1.8 were used for the following experiments.

RNA integrity was measured via a 2100 Bioanalyzer Instrument by RNA 6000 Nano Kit (Agilent Technologies, Santa Clara, CA, USA) and the samples with R.I.N. greater than 8 were used, while RNA quantification was performed using a Invitrogen™ Qubit™ RNA HS Assay Kit (Thermo Fisher Scientific, Waltham, MA, USA) via the Invitrogen™ Qubit™ 4 Fluorometer (Thermo Fisher Scientific, Waltham, MA, USA).

#### 4.3.2. Targeted RNA Sequencing

To analyze the differential expression level between olive cultivars, a targeted RNA sequencing approach, based on Illumina technology (Illumina^®^, San Diego, CA, USA) of the selected genes, was adopted.

The custom targeted panel of the 257 genes of interest, designed by Ilumina, included 48 and 112 genes involved in the phenylpropanoid and the fatty acid metabolism, respectively. Moreover, the gene set contains the other 97 genes playing a role in photoperception and the circadian clock. The genes were identified in the annotated genome of the Spanish cultivar Farga-Version 6 [18], the only available release in 2018 for cultivated olive, when the gene panel was designed and produced. The selection was performed via a manual search, using blastn and blastx tools, of the orthologous genes in different species. A first screening was carried out by blasting the *Arabidopsis thaliana* (Araport11) genes, and their function was characterized on the Farga genome (Oe6). The Farga transcripts (e.g., OE6A100286T1) with the highest blastn/blastx score was further blasted on different tree species to confirm the genes of interest’s sequence and function. At the 5′-end of each selected gene, specific primers were designed by Illumina. After several quality tests of annealing and amplification, a primer mix for the highly multiplexed polymerase chain reaction was suitable to be coupled with the AmpliSeq kit for the RNA-seq library preparation that was set up.

The original target transcript code from Farga (Oe6 release), the 5′ end and the 3′ end position of the left and right primers, together with the fully amplified fragment ranges, are reported in Appendix A. The sequenced and described genes have been reannotated; therefore, in all tables and figures are reported functional annotations fully updated for each gene via BLAST analysis.

RNA libraries were prepared via AmpliSeq^TM^ for Illumina (Illumina^®^, San Diego, CA, USA) according to the manufacturer’s protocol. Libraries were assessed via a 2100 Bioanalyzer Instrument using the DNA 1000 Kit (Agilent Technologies, Santa Clara, CA, USA) and DNA quantification was performed using the Invitrogen™ Qubit™ DNA HS Assay Kit (Thermo Fisher Scientific, Waltham, MA, USA) via the Invitrogen™ Qubit™ 4 Fluorometer (Thermo Fisher Scientific, Waltham, MA, USA). Paired-end (PE) complementary DNA (cDNA) libraries were sequenced using the MiniSEQ Instrument (Illumina^®^, San Diego, CA, USA).

The output is the same to a standard RNA-seq assay; thus, the analytic pipeline adopted was similar to a recent published study [60].

A reference genome has been used for mapping the reads to predict the genes’ (Oe6 release of ‘Farga’ by Cruz et al. [18].

Quality control checks of the raw sequence data coming from the Illumina sequencing were performed using FastQC v0.11.9 (Free Software Foundation, Inc., Boston, MA, USA). The adaptors were removed, and the low-quality regions were trimmed (Phred cut-off 20) by using Trimmomatic v0.39 (Free Software Foundation, Inc., Boston, MA, USA). The reads were aligned on the olive tree genome (cultivar Farga, release Oe6) via the aligned functions and the featureCounts in the Rsubread Bioconductor package v2.12.3. They were then filtered via expression, ruling out any transcript with an abundance less than 10, normalized using the TMM (Trimmed Mean of M Values), and finally analyzed for the differential expression using quasi-likelihood methods in the edgeR Bioconductor package v3.17.

### 4.4. Metabolite Analysis

#### 4.4.1. Phenolic Compounds

Phenolic extraction from olive drupes was achieved by using a hydroalcoholic solution of water/methanol (*v*/*v* 80:20). Briefly, in a 50 mL volume test tube containing 10 mL of the extraction solvent, 10 g of olives were reduced to a pulp by an ultra-turrax system at 8000 rpm for 1 min. To maximize the extraction process, the solution was kept under shaking in an ultrasonic bath in the darkness for 20 min. A centrifugation at 5000 rpm for 25 min at 10 °C allowed us to recover the supernatant that, after filtering using a 0.45 µm PVDF filter, was analyzed via HPLC-MS/MS.

##### HPLC–MS/MS Analysis

An HPLC 1200 series instrument (Agilent Technologies, Santa Clara, CA, USA) equipped with an Eclipse XDB-C8-A HPLC column (5 µm particle size, 150 mm length and 4.6 mm i.d.) was used to obtain the chromatographic separation of the phenolic compounds into the hydroalcoholic extracts. The binary mobile phase was composed via a 0.1% aqueous formic acid (A) and methanol (B), and the gradient was programmed to increase in B from 5% to 100% in 15 min, holding for 5 min and ramping down to the original composition (95% A and 5% B) in five minutes. The total elution time was 25 min per injection. The flow rate and the injection volume were 250 µL min^−1^ and µL, respectively. The identification of the phenolic compounds eluted was possible via a MSD Sciex Applied Biosystem API 4000 Q-Trap mass spectrometer. The analyses were conducted in negative ion mode using multiple reaction monitoring (MRM). The LC–MS experimental conditions were: ionspray voltage (IS) −4500 V; curtain gas 22 psi; temperature 350 °C; ion source gas (1) 35 psi; ion source gas (2) 45 psi; collision gas thickness (CAD) medium; and the entrance potential (EP), declustering potential (DP), entrance collision energy (CE) and exit collision energy (CXP) were optimized for each transition monitored.

##### Quantitative Analysis

Sigma–Aldrich (Riedel-de Haën, Laborchemikalien, Seelze, Germany) and Extrasynthese (Nord B.P 62 69726 Genay Cedex, France) provided the standards used for the analyses. Methanol and formic acid were LC/MS grade and purchased from VWR International; the aqueous solutions were prepared using ultrapure water, with a resistivity of 18.2MO cm and obtained from a Milli-Q plus system (Millipore, Bedford, MA, USA). Quantitative analyses were performed via the external calibration curves built using a least squares linear regression analysis. For this purpose, the standard stock solutions of caffeic acid (Caf), vanillic acid (Vco), o-cumaric acid (o-Cum), p-cumaric acid (p-Cum), apigenin (Ap), apigenin-7-O-glucoside (Ap7), diosmetin (Dio), hydroxytyrosol (HyTyr), tyrosol (Tyr), oleuropein (Olp), luteolin (Lut), verbascoside (Ver), luteolin-7-O-glucoside (Lu7), luteolin-4-O-glucoside (Lu4) rutin (Rut), and catechol (Cat) were dissolved in methanol and further diluted with 0.1% formic acid in water to obtain six calibration standards at concentrations in the range between 100 and 2000 µg ml^−1^. The correlation coefficients of the calibration curve ranged from 0.9994 to 0.9997. Each compound was monitored via the MRM mode which scans, on the third quadrupole, the main fragments of the deprotonated molecular ion [M-H]-1 produced in the first quadrupole. The analysis parameters, such as the equation for the external calibration curve, the correlation coefficient R^2^, the molecular ion [M-H]-1 monitored on the first quadrupole, and the major fragments monitored on the third quadrupole, for each phenolic compound, were analyzed via LC-MS/MS, as summarized in Appendix A.

#### 4.4.2. Fatty Acids

Fatty acid methyl esters (FAMEs) following the method described by Christie [61] were prepared. Briefly, 100 µL of a methanolic solution of potassium hydroxide (KOH) (2 N) and n-hexane were added to 0.15 g of oil, for reaching a volume of 1.5 mL. The resulting solution was shook vigorously for 5 min at room temperature. Afterwards, 0.2 µL of the supernatant was dissolved in n-hexane for a final volume of 2 mL and injected into a gas chromatographer (GC). The analyses were conducted by means of an Agilent GC (6890N) equipped with a capillary column SP-2340 (60 m × 0.25 mm i.d., 0.2 μm f.t., Supelco, Bellefonte, PA, USA) and a flame ionization detector (FID). Nitrogen was used as the carrier gas. The temperature of the column, injector, and detector were set at 180 °C, 230 °C, and 260 °C, respectively. The separation of the analytes was carried out by programming the temperature as follows: 110 °C held for 5 min, increase of 3 °C/min to 150 °C and held for 16 min, and increase of 4 °C/min to 230 °C and held for 27 min. Peaks were identified by comparing their retention times to those of the reference compounds. The fatty acid composition was expressed as each fatty acid relative % calculated using the internal standard normalization of the chromatographic peak area.

#### 4.4.3. Chemical Analyses of Fruit Anthocyanins’ Contents

Total monomeric anthocyanin concentration was determined essentially as described in Lee et al. [62], with some modifications. Briefly, 0.5 g of frozen olive powder was extracted with 2.5 mL of cold methanol containing 1% chloridric acid, followed by a 1 h incubation at 60 °C. Samples were then filtered over 0.2 μm of PTFE and the samples were monitored at 700 and 520 nm. We calculate the anthocyanin pigment concentration, which is expressed as cyanidin-3-rutinoside equivalents, as follows:$$Anthocyanin\;pigment\;(cyanidin\;3\text{-}rutinoside\;equivalents,\;mg/L)=\frac{A\times MW\times DF\times 10^{3}}{e\times l}$$
where A = (A520 nm–A700 nm); MW (molecular weight) = 631 g/mol for cyanidin-3-rutinoside (cyd-3-rut); DF = dilution factor established in D; l = pathlength in cm; e = 26,100 molar extinction coefficient, in L x mol^−1^ x cm^−1^, for cyd-3-rut; and 10^3^ = factor for conversion from g to mg. The data are summarized in Appendix A.

### 4.5. Bioinformatics Analysis

The raw reads were processed following the pipeline reported in Salimonti et al. [60] and archived in the NCBI SRA database under accession number PRJNA906028.

Several intra- and inter-cultivar pair comparisons were performed considering the ripening stages (T and R) within the same cultivar (CdN T vs. CdN R, Ru T vs. Ru R, and Sa T vs. Sa R) and the same ripening stage between two different cultivars (CdN T vs. Sa T, CdN R vs. Sa R, Ru T vs. Sa T, and Ru R vs. Sa R).

Differential expressed genes (DEGs) were obtained by the filtering process, taking into account only the transcripts with a fold Change (FC) > 2 or < −2 and a FDR p-adjusted value < 0.05 were considered significant.

A MDS (Multi-Dimensional Scale) analysis was carried out on the normalized and filtered gene counts via the plotMDS R function (limma package) to depict a principal coordinate analysis (PCoA) plot.

The average of the metabolites collected from each treatment (three cultivars and two ripening stages) were subjected to ANOVA, t test, and Tukey’s pairwise comparisons and box plots were produced using PAST software (https://www.nhm.uio.no/english/research/resources/past/, accessed on 21 September 2023). Comparison analysis between the DEGs and Differential Accumulated Metabolites (DAMs), applying the univariate correlation analysis (Pearson correlation coefficient-PCC), was performed using PAST software and the PCC values were visualized using the conditional formatting function of Excel software v2308 (Microsoft Corporation, Redmond, WA, USA).

## 5. Conclusions

Our study evidenced a potential involvement of photoreceptors and circadian clock machinery in the production of photoprotective pigments, such as the anthocyanins, and lipid modification. The key role of transcriptional regulation in the expression of many genes involved in the FAs’ and phenylpropanoids’ pathways in olive was also highlighted.

Transcriptome- and metabolome-integrated analyses on the contrasting genotypes for the anthocyanins’ and FAs’ content provided novel insights for understanding the key genes which may determine the formation of these metabolites. The candidate genes identified in this study might provide a basis for future functional genomics research aimed at the optimization of flavonoid and oleic acid content for increasing the oil’s nutraceutical value. At the same time, the interesting results obtained by using the targeted RNA-sequencing approach will encourage the development of new gene panels, promoting a more intensive use of this technique for differential expression analysis in olive.

## Figures and Tables

**Figure 1 ijms-24-14448-f001:**
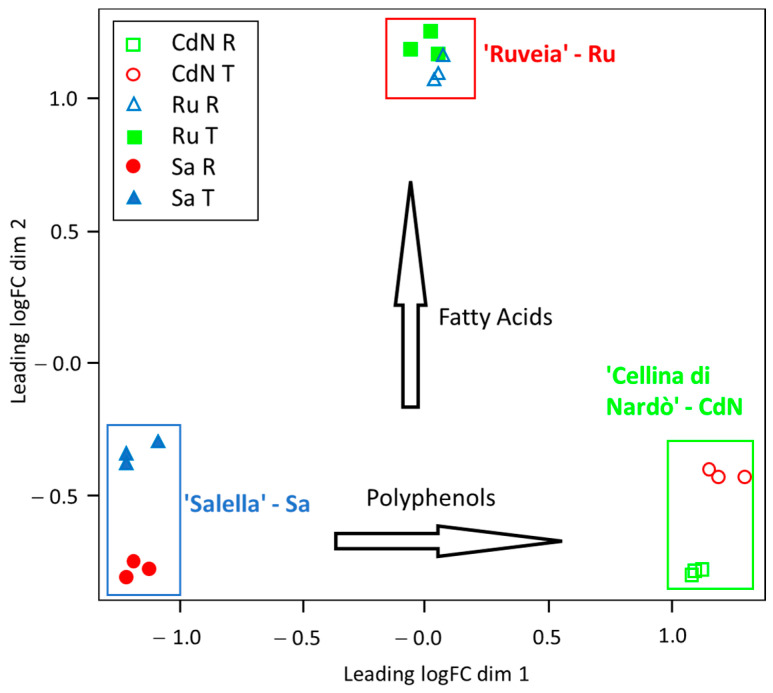
Principal coordinate (PCoA) plot shows MDS (MultiDimensional Scale) analysis on the previously normalized and filtered gene counts. Data refers to drupes from cv. Cellina di Nardò (CdN), Ruveia (Ru), and Salella (Sa) at Turning (T) and Ripe (R) stages. The major component of variance (x axis) significantly contributed to polyphenols’ variability that separated CdN and Sa cultivars. The y axis showed the variance for FAs able to distinguish R cultivar from the others.

**Figure 2 ijms-24-14448-f002:**
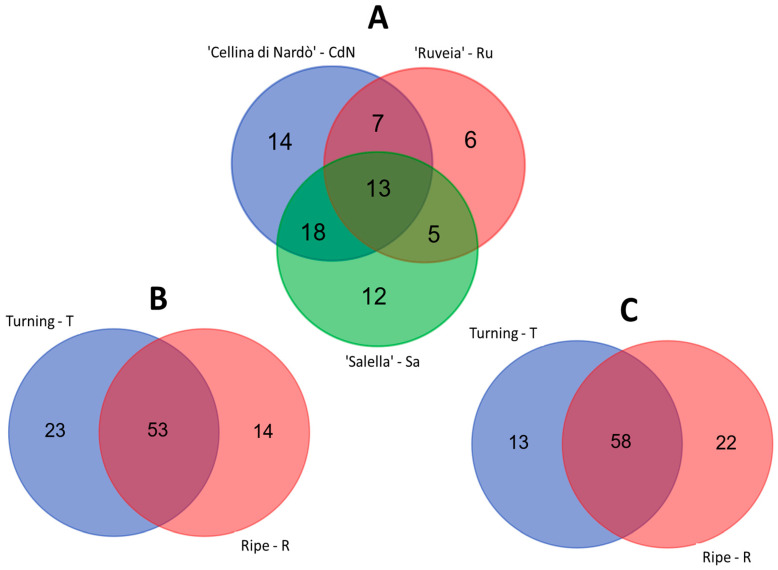
Venn diagram of DEGs extracted from (**A**) Turning (T) vs. Ripe (R) comparison for each cultivar, from (**B**) ‘Cellina di Nardò’ (CdN) vs. ‘Salella’ (Sa) comparison at two ripening stages, Turning (T) and Ripe (R), and from (**C**) ‘Ruveia’ (Ru) vs. ‘Salella’ (Sa) comparisons at two ripening stages, Turning (T) and Ripe (R).

**Figure 3 ijms-24-14448-f003:**
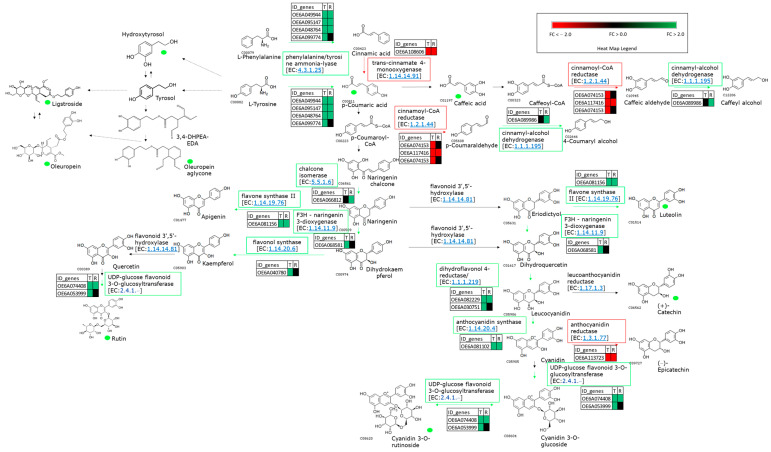
Representation of part of phenylpropanoids and oleuropein biosynthetic pathway in olive fruits. For each enzymatic step, rectangular boxes outlined in green and red indicate up- and down-regulated genes in ‘Cellina di Nardò’ vs. ‘Salella’ comparison, respectively. Heat map of transcript expression for each DEG was reported. The green dots indicate the metabolites more accumulated in ‘Cellina di Nardò’ with respect to ‘Salella’. The solid arrows indicate the consequential processes, and the dashed arrows represent those processed with unknown intermediates. Fold Change (FC).

**Figure 4 ijms-24-14448-f004:**
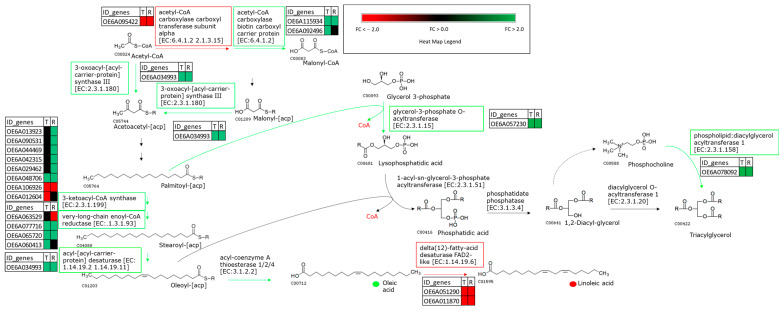
Representation of Fatty Acids’ and TAGs’ biosynthetic pathway in olive fruits. For each enzymatic step, rectangular boxes outlined in green and red indicate up- and down-regulated genes in ‘Ruveia’ vs. ‘Salella’ comparison, respectively. Heat map of transcript expression for each DEG was reported. The green and red dots indicate the metabolites more and less accumulated in ‘Ruveia respect to ‘Salella’, respectively. Fold Change (FC).

**Figure 5 ijms-24-14448-f005:**
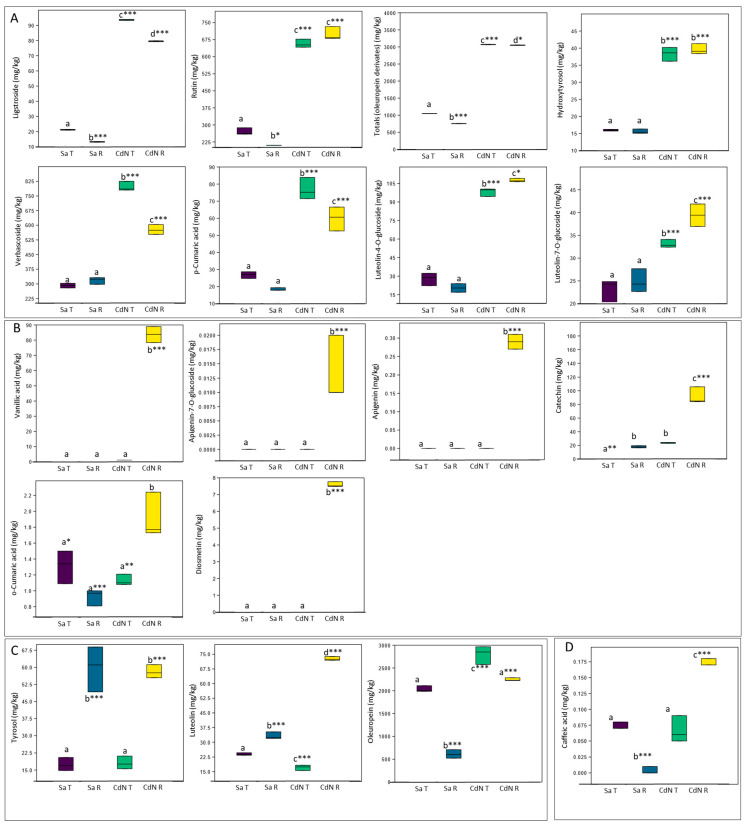
Box plot charts with quantitative levels of phenolic compounds in ‘Cellina di Nardò’ (CdN) and ‘Salella’ (Sa) at Turning (T) and Ripe (R) fruit stages. (**A**) Metabolites strongly accumulated in CdN in both ripening stages; (**B**) metabolites strongly accumulated in CdN at R stage; (**C**) metabolites showing a similar accumulation pattern mediated via ripening process in both cultivars; (**D**) metabolite showing a different accumulation pattern mediated via ripening process between the cultivars. For each metabolite, different letters indicate significant differences among mean values (* *p* < 0.05; ** *p* < 0.01; *** *p* < 0.001).

**Figure 6 ijms-24-14448-f006:**
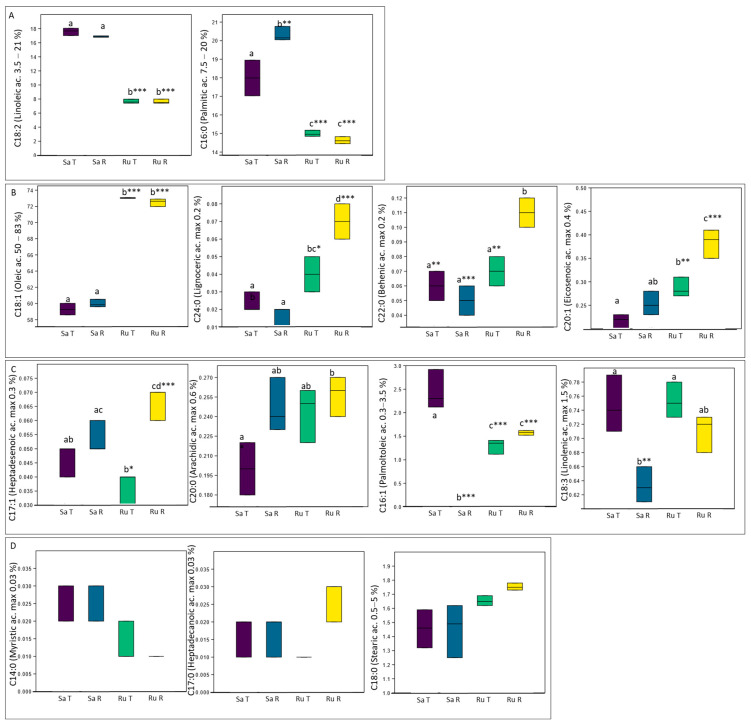
Box plot charts with FAs’ percentage in ‘Ruveia’ (Ru) and ‘Salella’ (Sa) at Turning (T) and Ripe (R) fruit stages. (**A**) Metabolites strongly accumulated in Sa in both ripening stages; (**B**) metabolites strongly accumulated in Ru in both ripening stages; (**C**) metabolites without specific expression pattern; (**D**) metabolites not differentially expressed. For each metabolite, different letters indicate significant differences among mean values (* *p* < 0.05; ** *p* < 0.01; *** *p* < 0.001).

**Figure 7 ijms-24-14448-f007:**
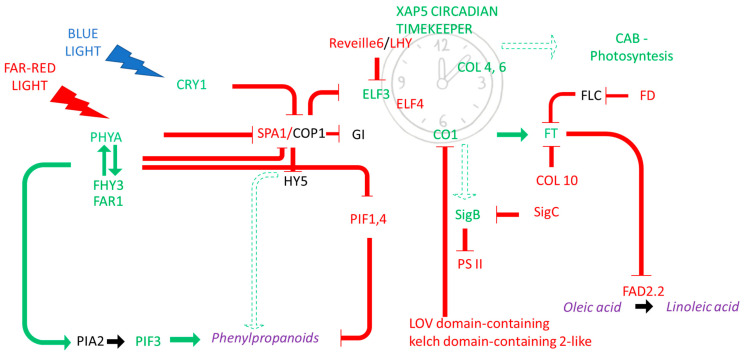
A schematic model summarizing the light-mediated genes and the molecular mechanisms involved in the phenylpropanoids and FAs biosynthesis regulation. Blue and far-red light would promote the production of photoprotective pigments, as anthocyanins, by the increase in *CRY1* and *PHYA* transcript levels and the regulation of clock machinery. The concurrent *PHYA*, *FHY3*, *FAR1,* and *PIF3* up-regulation and *SPA1* and *PIF4* down-regulation in both ‘Cellina di Nardò’ and ‘Ruveia’ is coherent with an increase in phenylpropanoids levels mediated via far-red light. Similarly, the *CRY1* up-regulation seems to limit the repressor effect on the transcription factors *SPA1* and *HY5*. Circadian clock genes would promote the *CONSTANS* florigen gene, which in turn would promote the *FT* gene, further influenced by decreased levels of *FD* observed only in ‘Ruveia’ and *COL10*. A repressor role for this transcription factor is conceivable in the regulation of *FAD2.2*. The green arrows and red lines represent the stimulatory and inhibitory effect, respectively. The up- and down-regulated genes are marked in green and red, respectively. The solid arrows indicate the consequential processes, and the dashed arrows represent those processed with unknown intermediates.

**Table 1 ijms-24-14448-t001:** Oleic and Linoleic acid percentage calculated via integration of unbiased signals in the ^1^H ZG NMR spectra [30] obtained using quantitative conditions (integrals values inaccuracy < 2.0% [31]. Phenolic compounds (* integral values of selected unbiased of signals in ^1^H NMR NOESYGPPS NMR spectra for comparison porpoises, see Section 4). Analysis performed on fruit samples collected during 2018 olive oil season, as reported in Section 4.2.

	% Oleic Acid			% Linoleic Acid			Phenolic Compounds *		
Cultivar	Green	Turning	Ripe	Green	Turning	Ripe	Green	Turning	Ripe
Cellina di Nardò	74.92	71.35	67.36	6.88	8.92	11.18	0.27	0.23	0.21
Ruveia	77.92	76.36	76.10	5.06	6.45	7.10	0.16	0.13	0.10
Salella	62.51	60.14	58.00	14.57	16.32	17.95	0.12	0.11	0.11

## Data Availability

The data supporting the conclusions of this article (raw RNA-Seq reads) are available in the National Center for Biotechnology Information (NCBI) Sequence Read Archive (SRA): PRJNA906028. The Appendix A associated with this article have been uploaded to the submission system.

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
