# Peer review of "Combined Transcriptomic and Metabolomic Approach Revealed a Relationship between Light Control, Photoprotective Pigments, and Lipid Biosynthesis in Olives"

_ijms, 2023, doi:10.3390/ijms241914448_

Round 1

Reviewer 1 Report

In the article “Combined transcriptomic and metabolomic approach revealed a relationship between light control, photoprotective pigments, and lipid biosynthesis in olive” The mechanisms of light-regulated metabolite synthesis and gene expression in 3 olive cultivars were studied. Light quality and quantity play an important role in plant growth and fruit quality formation. However, the research content in this paper is not innovative enough to be published in the International Journal of Molecular Sciences. In addition, the following problems still exist in this manuscript, and I hope the author can revise it carefully.

The introduction should be a mini overview of the research topic of this paper, but it is too redundant, blindly piling up literature, and the logic is relatively confused.

The result section is also redundant (not concise). In the results, some genes or transcripts were identified to be related to the photosystem. However, as far as I know, all green plants can annotate the genes related to photosynthesis after transcriptome sequencing, because the plants themselves have the photosynthesis ability. Therefore, I think the author's statement is not novelty.

In the results section, e.g. 2.1.2. and 2.1.3., the authors compare transcriptome data from different cultivars. Suppose the genetic background of the three varieties is very different. In that case, it is not suitable for direct comparison, due to the author did not introduce the genetic background of the three varieties.

The logic of the discussion was chaotic, for example, the subtitle "3.1. Maturation-related variations of phenylpropanoids and FAs biosynthetic pathways in different genotypes" does not match the content. They discuss chlorophyll and photosynthesis at great length in this part. Should there be a separate subheading for this section?

The subject of this manuscript is to study light-controlled metabolite and gene expression changes in olives. However, the method has nothing about the light quality and other treatments. The mere comparison of different olive varieties and two maturity stages, in my opinion, these results absolutely inadequate to elucidate the research question. Moreover, the authors are not clear about the genetic background, sampling details, and sample sources of the three different olive varieties.

To sum up, I think this article still needs great revision. It is not yet enough for publication in the International Journal of Molecular Sciences.

Author Response

In the article “Combined transcriptomic and metabolomic approach revealed a relationship between light control, photoprotective pigments, and lipid biosynthesis in olive” The mechanisms of light-regulated metabolite synthesis and gene expression in 3 olive cultivars were studied. Light quality and quantity play an important role in plant growth and fruit quality formation. However, the research content in this paper is not innovative enough to be published in the International Journal of Molecular Sciences. In addition, the following problems still exist in this manuscript, and I hope the author can revise it carefully.

First of all, I am so grateful for clarifying me some aspects to be improved in order to make the manuscript more readable. Quality and light play an important role in many vital processes for plants. For this reason, we investigated these dynamics, for the first time, in the olive tree. At the same time, we have been studied more targeted experiments in growth chamber to deep characterize the effects of light in terms of both intensity (length of day and different time of the day) and quality (exposure to specific light spectra). Our approach is a novelty because it is the first attempt in the olive tree to elucidate the key role of some genes (targeted-RNA sequencing, another important novel approach), already known in other species in the mediation of light, the regulation of important olive expressed traits, such as  the accumulation of phenols/phenylpropanoids and of FAs/TAGs.

The introduction should be a mini overview of the research topic of this paper, but it is too redundant, blindly piling up literature, and the logic is relatively confused.

Thank you for this suggestion, we revised the introduction making it more concise. Please, find all changes highlighted in yellow in the main text.

The result section is also redundant (not concise). In the results, some genes or transcripts were identified to be related to the photosystem. However, as far as I know, all green plants can annotate the genes related to photosynthesis after transcriptome sequencing, because the plants themselves have the photosynthesis ability. Therefore, I think the author's statement is not novelty.

Thank you also for this suggestion, we revised the result section considering your comments, but I would like to clarify some aspects related to our results. The aim of our research was not to identify an atlas of expressed genes, indeed, it is not based on a traditional shotgun transcriptomics approach, instead of but a targeted approach (targeted-RNA) on some genes selected for their involvement in the pathways of our interest was adopted. In particular, in the genes panel we included those related to the photosystems finely regulated during fruit ripening with the typical transition ‘chloroplasts to chromoplasts’ expected during the veraison of olives ‘green to more pigmented and ripe stages’. These genes are also markers of plant responses to light and together with others already known for their direct and indirect involvement in the response to light stimuli, allowed us to verify the different photo-perceptive ability of our genotypes grown in the same environmental conditions.

In the results section, e.g. 2.1.2. and 2.1.3., the authors compare transcriptome data from different cultivars. Suppose the genetic background of the three varieties is very different. In that case, it is not suitable for direct comparison, due to the author did not introduce the genetic background of the three varieties.

Thank for your comments, the experimental design included different comparisons, the same cultivar at different stages and comparisons between CVs that is the main stream of a transcriptomics comparative analysis. The comparison between CVs, if correctly upstream phenotyped, is currently considered the most practicable method to carry out studies to elucidate and describe the involvement of specific genes in specific metabolic processes. Moreover, the three CVs belong to the same botanical variety (O. europaea subs. europaea var. europaea). The cultivated olive germplasm is not the result of large genetic improvement programs, as demonstrated by the highly restricted genetic base observing in our studies, currently underway, based on whole genome genotyping on the same reference genome. Finally, the choice of using the Turning stage as Time 0 (T0) allowed us to identify a developmental stage largely unchanged among the tree cultivars, thanks to the use of an index based on average coloring of olive fruits (Jaén index)

Below are some of the many articles that base similar studies on cultivar comparisons.

Dabbou S, Dabbou S, Selvaggini R, Urbani S, Taticchi A, Servili M, Hammami M. Comparison of the chemical composition and the organoleptic profile of virgin olive oil from two wild and two cultivated Tunisian Olea europaea. Chem Biodivers. 2011 Jan;8(1):189-202. doi: 10.1002/cbdv.201000086. PMID: 21259429.

Celenk S, Vatansever B. Assessment of heterogeneity of two cultivars of Olea europaea based on the study of their Ole e 1 protein content. Environ Sci Pollut Res Int. 2021 Feb 27;28(25):33545–56. doi: 10.1007/s11356-021-13122-2. Epub ahead of print. PMID: 33641102; PMCID: PMC7914038.

Iaria DL, Chiappetta A, Muzzalupo I. A De novo Transcriptomic Approach to Identify Flavonoids and Anthocyanins "Switch-Off" in Olive (Olea europaea L.) Drupes at Different Stages of Maturation. Front Plant Sci. 2016 Jan 19;6:1246. doi: 10.3389/fpls.2015.01246. PMID: 26834761; PMCID: PMC4717290.

Ferrari M, Muto A, Bruno L, Muzzalupo I, Chiappetta A. Modulation of Anthocyanin Biosynthesis-Related Genes during the Ripening of Olea europaea L. cvs Carolea and Tondina Drupes in Relation to Environmental Factors. Int J Mol Sci. 2023 May 15;24(10):8770. doi: 10.3390/ijms24108770. PMID: 37240115; PMCID: PMC10217892.

Alagna F, D'Agostino N, Torchia L, Servili M, Rao R, Pietrella M, Giuliano G, Chiusano ML, Baldoni L, Perrotta G. Comparative 454 pyrosequencing of transcripts from two olive genotypes during fruit development. BMC Genomics. 2009 Aug 26;10:399. doi: 10.1186/1471-2164-10-399. PMID: 19709400; PMCID: PMC2748093.

Hernández ML, Sicardo MD, Belaj A, Martínez-Rivas JM. The Oleic/Linoleic Acid Ratio in Olive (Olea europaea L.) Fruit Mesocarp Is Mainly Controlled by OeFAD2-2 and OeFAD2-5 Genes Together With the Different Specificity of Extraplastidial Acyltransferase Enzymes. Front Plant Sci. 2021 Mar 8;12:653997. doi: 10.3389/fpls.2021.653997. PMID: 33763103; PMCID: PMC7982730.

Yi Wu, Cheng-Wei Qiu, Fangbin Cao, Li Liu, Feibo Wu, Identification and characterization of long noncoding RNAs in two contrasting olive (Olea europaea L.) genotypes subjected to aluminum toxicity, Plant Physiology and Biochemistry, 2023, https://doi.org/10.1016/j.plaphy.2023.107906

Arias NS, Bucci SJ, Scholz FG, Goldstein G. Freezing avoidance by supercooling in Olea europaea cultivars: the role of apoplastic water, solute content and cell wall rigidity. Plant Cell Environ. 2015 Oct;38(10):2061-70. doi: 10.1111/pce.12529. Epub 2015 Apr 23. PMID: 25737264.

Petridis A, Therios I, Samouris G, Koundouras S, Giannakoula A. Effect of water deficit on leaf phenolic composition, gas exchange, oxidative damage and antioxidant activity of four Greek olive (Olea europaea L.) cultivars. Plant Physiol Biochem. 2012 Nov;60:1-11. doi: 10.1016/j.plaphy.2012.07.014. Epub 2012 Aug 2. PMID: 22885895.

The logic of the discussion was chaotic, for example, the subtitle "3.1. Maturation-related variations of phenylpropanoids and FAs biosynthetic pathways in different genotypes" does not match the content. They discuss chlorophyll and photosynthesis at great length in this part. Should there be a separate subheading for this section?

Thank you for this comment, we revised the discussion section considering your suggestions. In particular, we changed the subtitle of 3.1 section to clarify the correct meaning of our discussion. In this section, we confirmed the genes expression patterns known in olive about phenylpropanoids and FAs and in other species about light perception and photosynthetic processes. We think that this knowledge may strengthen the new findings argued in the next section of the discussion

The subject of this manuscript is to study light-controlled metabolite and gene expression changes in olives. However, the method has nothing about the light quality and other treatments. The mere comparison of different olive varieties and two maturity stages, in my opinion, these results absolutely inadequate to elucidate the research question. Moreover, the authors are not clear about the genetic background, sampling details, and sample sources of the three different olive varieties.

Thank you for this valuable comment. We would like to clarify that our experimental design included three CVs analyzed at different stages for their gene expression utilizing a targeted approach with a panel of specific genes. The results allowed us to identify some significant differences in several pathways among which the light perception correlated to important olive oil traits. But the objective of our study is not to clarify in detail all light-mediated mechanisms influencing metabolic processes in olive trees. As many similar approaches, our results were precisely defined, discussed and corroborated by many references that often demonstrated that our results were shared by other significant studies on olive tree and other plants. Further experiments are currently underway to better clarify the scientific aspects and surely functional studies will be planned to demonstrate the function of some identified genes. Otherwise, we believe that this study started to deepening the crucial role of photoperception in the modulation of important traits expressed by olive tree (accumulation of phenols and fatty acids for which the three cv have been deep phenotyped).

Reviewer 2 Report

The present work focused on the systemic analysis of relationship between accumulation of valuable lipids, secondary metabolites in three olive cultivars. An integrated approach including transcriptomics and metabolomics makes the work valuable both for practical use and interesting for fundamental researchers. The authors used modern methods of metabolomics and transcriptomics in manner suitable for goals were claimed. The authors summed up revealed data in the model of the metabolic and signaling mechanism.

This work looks as complete investigation. However, the manuscript raises few comments:

1. Illustrative material needs improvment to be more readable.

2. “Differential Expressed Metabolites (DEMs)” - is not good term, because, metabolites are not encoded and expressing, it would be better use “differently accumulated metabolites (DAM)”.

Author Response

The present work focused on the systemic analysis of relationship between accumulation of valuable lipids, secondary metabolites in three olive cultivars. An integrated approach including transcriptomics and metabolomics makes the work valuable both for practical use and interesting for fundamental researchers. The authors used modern methods of metabolomics and transcriptomics in manner suitable for goals were claimed. The authors summed up revealed data in the model of the metabolic and signaling mechanism.

I'm so grateful for the positive review.

This work looks as complete investigation. However, the manuscript raises few comments:

  1. Illustrative material needs improvment to be more readable.

We improved the captures and the figures

  1. “Differential Expressed Metabolites (DEMs)” - is not good term, because, metabolites are not encoded and expressing, it would be better use “differently accumulated metabolites (DAM)”.

We replaced DEMs with DAMs

Please, find all changes highlighted in yellow in the main text.

Round 2

Reviewer 1 Report

The author has answered all questions I mentioned and revised most of the comments in the manuscript. However, the author remains unchanged in the introduction and discussion sections, the introduction remains lengthy and the discussion is illegible.

Author Response

Thank you to have changed your mind about the experimental design and about the goodness of our work. Please, find all changes highlighted in green (last revision) and in yellow (previous revision) in the main text.

In order to make it more legible, we have further reduced the introduction by eliminating the redundant parts. On the other hand, we believe that the introduction must necessarily contain a description of all aspects discussed in our study: biophenols and FAs associated to healthy properties of EVOO (olive oil), photoperception and ripening representing mediator and crucial step in the regulation of these pathways, respectively, and finally the adopted omics approach.

In the light of your first revision pointing to the quality of subsection 3.1, we maintained the proposed title in the previous revision, and we completely revised the text. In order to make it clearer and more legible, we revised main parts of the text to make it more coherent and understandable for the readers. Moreover, we emphasized the substantial objective of the whole subsection which aims to confirm “the known genes expression patterns of phenylpropanoids, FAs, light perception and photosynthetic processes. In our view, this knowledge may strengthen the strictly genotype-dependent findings, argued in the next section of the discussion 3.2”